# The changing impact of the active job openings-to-applicants ratio (AJOAR) on ambulance dispatches during deflation: A longitudinal ecological study

Yohei Kamikawa[1]*, Nao Hanaki[2]

1 Department of Emergency Medicine, University of Fukui Hospital, Fukui, Japan, 2 Department of Public Health, Osaka University Graduate School of Medicine, Osaka, Japan

* fundarike.ykami@gmail.com

## Abstract

### Background

Frequent ambulance dispatches is a common challenge in developed countries. Several factors have been identified as contributing to increase in dispatches, but no stipulation has explained the particular shift observed in Japan since 1995. This study examined ambulance dispatches in view of changes in a macroeconomic indicator.

### Methods

This longitudinal ecological study covered all annual ambulance dispatch incidents in Japan between 1980 and 2021 (42 years). The regression model comprised the active job openings-to-applicants ratio during deflation, the active job openings-to-applicants ratio during inflation, aging population trend, and mean ambient temperature, with the Japanese total population as an offset variable.

### Results

There were a total of 177,042,244 ambulance dispatches during the study period. The active job openings-to-applicants ratio during deflation showed statistical significance in the regression analysis (generalized estimation equations estimate: 0.165, 95% confidence interval: 0.087 to 0.243) whereas the active job openings-to-applicants ratio during inflation did not (0.019, −0.021 to 0.059).

### Conclusion

The active job openings-to-applicants ratio during deflationary periods was associated with increased ambulance dispatches.

**Data availability statement:** The data extracted for this study are available from the Open Science Framework database (https://osf.io/rge3j/).

**Funding:** The author(s) received no specific funding for this work.

**Competing interests:** The authors have declared that no competing interests exist.

## Introduction

### Background

Ambulance dispatches increased over time in developed countries [1–3]. Calls for ambulances in England increased from 8.2 million in 2011 to 14 million in 2021 [2]. In Japan, dispatches rose from 5.7 million in 2011 to 6.2 million in 2021 [3], with nearly half of these cases involving non-urgent patients [4]. This surge contributes to Emergency Department (ED) overcrowding. Fatigue among emergency medical teams due to overburden by ambulance use leads to compromised quality of medical care and delayed treatment for urgent patients [4–6]. Moreover, the increase in ambulance use exacerbates the shortage of medical resources and soaring national medical care expenditure [4,6].

Studies have explored reasons for the increase in ambulance dispatch rate, identifying factors such as older population, mean atmospheric temperature, and macroeconomic status, as influential contributors [7–12]. Particularly, the active job openings-to-applicants ratio (AJOAR), one of the preeminent coincident macroeconomic indicators for national macroeconomic status [13], was found to play a key role in the fluctuation of ambulance dispatches in Japan [12]. Increased AJOAR correlated with increased dispatches, and vice versa, between 2003 and 2021, demonstrating characteristic changes in ambulance dispatch numbers that sharply declined from 2005 to 2009, increased between 2010 and 2019, and dropped again in 2020 [12]. Nevertheless, there has been no research to explain the marked difference before and after 1995, when steep increment and fluctuations began (S1 Fig) [14].

In the field of economics, long-lasting economic growth is known to offset adverse effects on public health that would have occurred during temporary economic growth [15]. From this perspective, the mild and monotonous dispatch rate increase before 1995 might be attributed to long-term economic prosperity, for Japan faced an 'economic bubble burst' in the early 1990s leading to a prolonged period of deflationary recession starting from 1995 (S1 Fig) [16].

We hypothesized that AJOAR's impact on ambulance dispatches might change depending on the long-term macroeconomic status (i.e., deflationary recession or inflationary prosperity).

### Objective and impact

This study aimed to find the contribution of AJOAR to the number of ambulance dispatches in Japan during deflationary and inflationary periods. If a correlation exists, the decision to implement societal and economic policies to prevent deflation may reduce ambulance dispatches.

## Methods

### Study design and setting

This is a longitudinal ecological study. We used data showing annual ambulance dispatch numbers in Japan, covering both road-ambulance and helicopters, between January 1st, 1980, and December 31st, 2021. Ambulance services are provided free

of charge to all residents throughout Japan. The prehospital emergency medical system is managed by local fire departments. There were 724 fire department headquarters, 5,302 teams, and 65,181 crew members as of 2021, the last year of the study period [14]. The total population of Japan was approximately 125,502,000 and there were 6,196,069 ambulance dispatches as of 2021, about 5 occurrences per 100 population a year [14,17]. The Research Ethics Committee of University of Fukui approved this protocol (number 20230016) and we used only publicly available data.

## Procedures

Annual data were used in this study because both deflation and inflation have an impact on an annual basis [18]. The variables for analyses were selected in accordance with a prior study [12]. The annual ambulance dispatch data were obtained as outcome variables from the database of the Current Status of Emergency and Rescue Services of the Fire and Disaster Management Agency [14]. Estimates of the number of older individuals (age 65 and older) and the total population were derived from the Population Estimates of the Statistics Bureau of the Ministry of Internal Affairs and Communications [17]. Mean ambient temperatures were collected from the Japan Meteorological Agency [19]. AJOAR data was derived from the Employment Referrals for General Workers of the Ministry of Health, Labour and Welfare [20]. The gross domestic product (GDP) deflators were obtained from the International Monetary Fund [21], and the Tokyo Stock Price Index (TOPIX) was retrieved from the Japan Exchange Group [22].

Older people were defined as those aged 65 years and above based on previous studies finding them to use ambulances twice more frequently compared to younger individuals, even in non-emergency situations [7–9]. For the mean temperatures, we constructed a dataset of national mean temperatures by averaging all prefectural mean temperatures. Since the trends in mean temperatures were similar between the national average and every prefectural average [19], we concluded that national average could adequately represent the trend. Additionally, it has been shown that the number of ambulance dispatches is the lowest when the temperature is 22.5°C, increasing as it deviates from that point in a U-shaped relationship [10]. To address this U-shaped relationship in linear regressions, the data needed to be split into two portions; an upswing effect and a downswing effect. Therefore, a new dataset was created, setting data as 0 for mean temperatures below 22.5°C and as the mean temperature minus 22.5 for mean temperatures above 22.5°C, termed the "high-temperature effect." Conversely, another dataset was created, where data were set as 0 for mean temperatures above 22.5°C and as the mean temperature minus 22.5 for mean temperatures below 22.5°C, termed the "low-temperature effect." However, no data on the national average temperatures exceeded 22.5°C, so only the low-temperature effect was incorporated as the mean temperature variable. Admittedly, it may be apparent that mean temperature cannot act as a confounding variable when analyzing annual trends, as the seasonal effects on ambulance dispatches—often represented by mean temperature—are not reflected in such analyses. However, since this study follows the methodology of a previous study that demonstrated the significant impact of mean temperature [12], excluding this variable would seem overly arbitrary. Therefore, mean temperature was retained as a variable in this study.

The GDP deflator, calculated as (Nominal GDP/Real GDP)×100, is an economic index that represents the degree of price fluctuation [23]. Rising during inflation and falling during deflation, it is considered a reliable indicator of overall economic activity [24]. In this study, deflation was defined as a decrease in the GDP deflator compared to the previous year, while inflation was defined as an increase in the GDP deflator compared to the previous year. However, to capture long-term trends, single-year deflation within a sustained inflationary trend, where inflation continues for consecutive years, was considered an outlier and treated as an inflation year. Similarly, single-year inflation within a deflationary trend was treated as a deflation year. Subsequently, the dataset for "AJOAR during deflation" was constructed by setting the values to 0 for inflationary years and leaving them unchanged for deflationary years. Similarly, the dataset for "AJOAR during inflation" was created by setting the values to 0 for deflationary years and leaving them unchanged for inflationary years.

Since the annual data on the GDP deflator were only available from 1980 and ambulance dispatch data were available until 2021, the period from 1980 to 2021 was covered for analyses. All data was accessed on December 28, 2023. The data extracted for this study are available from the Open Science Framework database [25].

## Other factors

In an ecological study, variables should be selected to align with the group-level of the target outcomes [26]. While various factors have been reported in the past to be relevant to the increase in ambulance dispatches, most of them do not reflect national annual trends—except for the older population, mean temperature, and AJOAR. Instead, they can be broadly categorized into two major categories: regional and individual. Regional factors encompass population density [27,28], exciting events [29], air pollutants [30], limited hospital accessibility [7–9,31,32], and excessive accessibility to hospitals [7,32,33]. Individual factors include a lack of knowledge about health and medical resources [8,9,31], low income [8,9,31,32], a limited educational background [31], a sense of isolation [7,8,31], anxiety arising from a sense of powerlessness [9], and situations that promote risk overestimation (e.g., heavy responsibility, excessive advice from outsiders) [31]. Describing these factors as nationwide trends is challenging due to wide variations among regions or individuals, making it impossible to implement comprehensive measures. In contrast, macroeconomic indicators, older population, and temperature reflect nationwide trends, with macroeconomic indicators being particularly subject to government intervention. Notably, a previous study indicated that a simple model using older population, temperature, and AJOAR can estimate ambulance dispatch trends without accounting for those regional and individual factors between 2003 and 2021 in Japan [12]. Other studies have also employed similarly simple models [10,11].

Additionally, to ensure the reliability of the regression analysis, the number of explanatory variables should be limited to one-tenth of the outcome data points [34,35]. Given that this study includes only 42 annual outcome data points, a maximum of four variables can be used in the regression.

Therefore, this study did not consider those regional and individual factors.

## Statistical analyses

The characteristics of the outcome and variables were summarized in medians and interquartile ranges (IQRs)
For the main analysis, we included four explanatory variables in the regression to explain the outcome of the number of ambulance dispatches: the older population, the low-temperature effect, AJOAR during deflation, and AJOAR during inflation. Additionally, the total population was included as an offset variable. The generalized estimation equation (GEE) using a first-order autoregressive covariance structure and assuming a log link Poisson distribution were performed for the regression model, and robust variance estimators were used for final consequences. A Pearson's $\chi^2$ test was conducted as a goodness-of-fit test. In this test, the model was deemed to fit the observed trend if the p-value was 0.05 or higher [36]. As a sensitivity analysis, similar analyses were conducted using a model in which AJOAR in the main analysis model was replaced with TOPIX.

In the secondary analysis, similar analyses were performed using the explanatory variables of the older population, low-temperature effect, and unadjusted AJOAR, with the total population as an offset variable, based on the previous study that demonstrated the number of ambulance dispatches in Japan between 2003 and 2021 [12]. Subsequently, the quasi-likelihood under the independence model information criterions (QICs) were compared between the main and secondary analyses. A lower QIC indicates a better-fitting model [37].

Additionally, graphs from the main and secondary analyses were visually compared to the actual dispatch trend to validate the analyses outcomes. If the model of main analysis exhibited the better fit as expected, a recalculation was performed by assigning the regression coefficient of AJOAR during inflation to the coefficient of AJOAR during deflation in the GEE of the main analysis. This was done to simulate the trend in ambulance dispatches in a scenario where deflationary

periods were absent. The simulated estimates were then compared to the actual dispatch trend visually through a graph and by calculating the cumulative difference between them.

For reference, major socioeconomic events—such as the economic bubble burst, the global financial crisis, the Great East Japan Earthquake, the introduction of the additional fee for a first-time patient without a referral, and the COVID-19 pandemic—were included as variables in the main model. Then, we conducted a similar regression analysis and validated the results by generating a corresponding graph.

The research hypotheses were that the contribution of AJOAR during deflation was greater than that of AJOAR during inflation, that the sensitivity analysis showed similar results to the main analysis, and that the model of main analysis would show the better fit compared to the secondary analysis. Furthermore, the simulated number of ambulance dispatches would be fewer than the actual counts if there had been no deflationary periods. If, contrary to our expectation, the contribution of AJOAR did not change between deflation and inflation, the regression coefficients for AJOAR during deflation and inflation would show no difference in the main analysis.

The R software (version 4.4.1; The R Foundation, Vienna, Austria) was used for these analyses.

## Results

### Characteristics of study data

There were a total of 177,042,244 ambulance dispatches during the 42-year study period. The median values for this period were as follows: the annual number of ambulance dispatches, 4,290,101 (IQR 2,781,025–5,646,662); older population, 22,455,000 (IQR 15,091,500–29,685,000); low-temperature effect, –7.15 (IQR –7.56 to –6.68); AJOAR, 0.76 (IQR 0.63 to 1.09); AJOAR during deflation, 0 (IQR 0 to 0.64); AJOAR during inflation, 0.62 (IQR 0 to 1.09); total population, 126,513,500 (IQR 123,733,625–127,468,000); TOPIX 1423.41, (IQR 943.06 to 1673.24); TOPIX during deflation, 0 (IQR 0 to 1076.17); and TOPIX during inflation, 582.02 (IQR 0 to 1554.10). However, outliers in 1987, 1997, 2017, and 2021 were adjusted when constructing the dataset for AJOAR during deflation, AJOAR during inflation, TOPIX during deflation, and TOPIX during inflation. No data were missing for any variables.

### Main results

In the GEE of main analysis, AJOAR during deflation significantly contributed to the number of ambulance dispatches (GEE estimate: 0.165, 95% confidence interval [CI]: 0.087 to 0.243). In contrast, the GEE estimate of AJOAR during inflation was approximately one tenth of that during deflation, and moreover, did not significantly contribute to the number of dispatches (0.019, 95% CI: −0.021 to 0.059). While the older population showed a significant contribution (0.400, 95% CI: 0.381 to 0.418), the low-temperature effect did not (0.021, 95% CI: −0.006 to 0.048). The goodness-of-fit test demonstrated that the model fit was adequate ($\chi^2/df = 1.02$, $p = 0.20$).

The sensitivity analysis, conducted using TOPIX in place of AJOAR, yielded results consistent with the main analysis. The GEE estimate of TOPIX during deflation was 0.000067 (95% CI 0.000021 to 0.000113), which was statistically significant, while the estimate during inflation was not statistically significant (−0.000015, 95% CI −0.000046 to 0.000016). According to the goodness-of-fit test, the model showed an acceptable fit ($\chi^2/df = 1.02$, $p = 0.20$). A detailed summary of the sensitivity analysis results can be found in S1 Table.

In the secondary analysis, AJOAR showed significant but negative contribution (−0.070, 95% CI: −0.114 to −0.026). No evidence of poor model fit was identified in the goodness-of-fit test ($\chi^2/df = 1.02$, $p = 0.20$). The QIC of main analysis was lower than that of secondary analysis, indicating the former was better suited for explaining ambulance dispatch trends (−5,070,716,387 and −5,070,350,664, respectively). The results of GEE analyses are presented in Table 1.

When comparing the graphs, the main model generally aligned well with the actual trend, reproducing a mild increase from 1980 to 1994, a subsequent steep incline, a sharp decline around 2007, and a following increase. However, the

**Table 1. Results of the main and secondary analyses for the annual ambulance dispatches.**

| | Main analysis GEE estimate (95% CI) | Secondary analysis GEE estimate (95% CI) |
|---|---|---|
| AJOAR | | −0.070 (−0.114 to −0.026) * |
| AJOAR during deflation | 0.165 (0.087 to 0.243) * | |
| AJOAR during inflation | 0.019 (−0.021 to 0.059) | |
| Older population, $10^7$ people | 0.400 (0.381 to 0.418) * | 0.412 (0.383 to 0.441) * |
| Low-temperature effect | 0.021 (−0.006 to 0.048) | 0.057 (0.022 to 0.092) * |
| QIC | −5,070,716,387 | −5,070,350,664 |

GEE: generalized estimation equation; CI: confidence interval; AJOAR: active job openings-to-applicants ratio; QIC: quasi-likelihood under the independence model criterion. * $P < .05$. The regression model of main analysis consists of the older population, mean temperature, AJOAR during deflation, and AJOAR during inflation, including the total population as an offset variable. The regression model of secondary analysis consists of the older population, mean temperature, and AJOAR, including the total population as an offset variable.

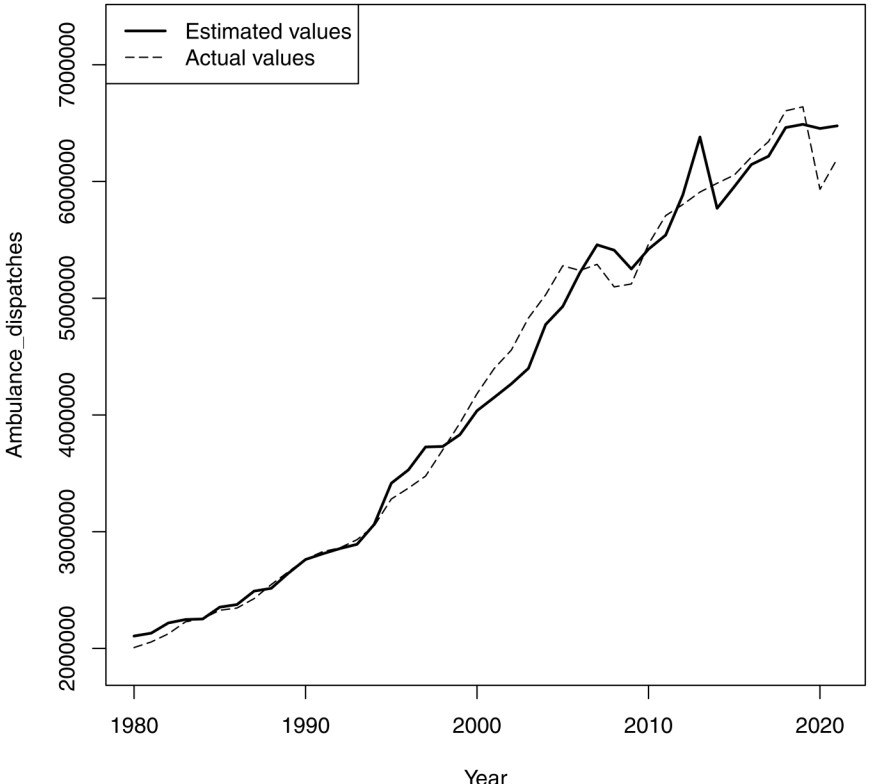

**Fig 1. Ambulance dispatches in Japan and the regression estimates of main analysis.** The regression model consists of the older population, mean temperature, active job openings-to-applicants ratio during deflation, and active job openings-to-applicants ratio during inflation, including the total population as an offset variable.

fitness around 2013 was uncertain (Fig 1). The model of the sensitivity analysis showed a graph similar to that of the main analysis (S2 Fig). In contrast, the model of secondary analysis demonstrated a relatively monotonous increase throughout the study period against the actual trend (Fig 2). This visual comparison was consistent with the statistical results that demonstrated the better fit for the model of main analysis than the secondary analysis.

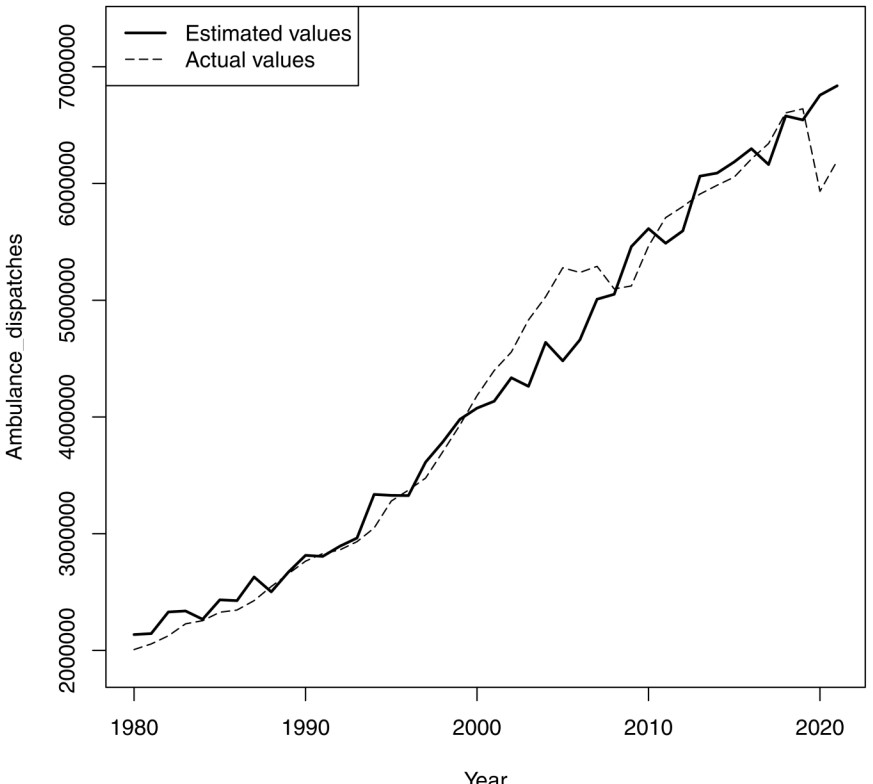

**Fig 2. Ambulance dispatches in Japan and the regression model of secondary analysis.** The regression model consists of the older population, mean temperature, and active job openings-to-applicants ratio, including the total population as an offset variable.

The simulated estimates of the ambulance dispatch trend, recalculated by assigning the regression coefficient for AJOAR during inflation (0.019) to the coefficient for AJOAR during deflation (0.165) in the GEE of main analysis, were generally fewer than the actual counts, especially from 1995 to 2013, which was almost the deflationary period (Fig 3). The cumulative difference over the study period was 9,026,431, while that in the deflationary period from 1995 to 2013 was 9,322,552.

When major socioeconomic events were included as variables, the economic bubble burst, the global financial crisis, and the COVID-19 pandemic demonstrated statistical significance (−0.040 [−0.063 to −0.017], −0.066 [−0.100 to −0.031], and −0.088 [−0.120 to −0.056], respectively), alongside AJOAR during deflation and the older population. In contrast, the Great East Japan Earthquake and the introduction of the additional fee for a first-time patient without a referral did not show statistical significance. The goodness-of-fit test indicated no issues with the model fit ($\chi^2$/df = 1.02, p = 0.20). The QIC of the model was −5,070,814,552, and the graph closely aligned with the actual trend of ambulance dispatches, indicating a better fit compared to the main model. These results were presented in S2 Table and S3 Fig.

## Discussion

### Findings

This study discovered that AJOAR during deflationary periods was associated with an increase in ambulance dispatches, whereas during inflationary periods, it was not associated with the number of ambulance dispatches. Ultimately, considering macroeconomic indicators' contribution improved the accuracy of ambulance dispatch estimation.

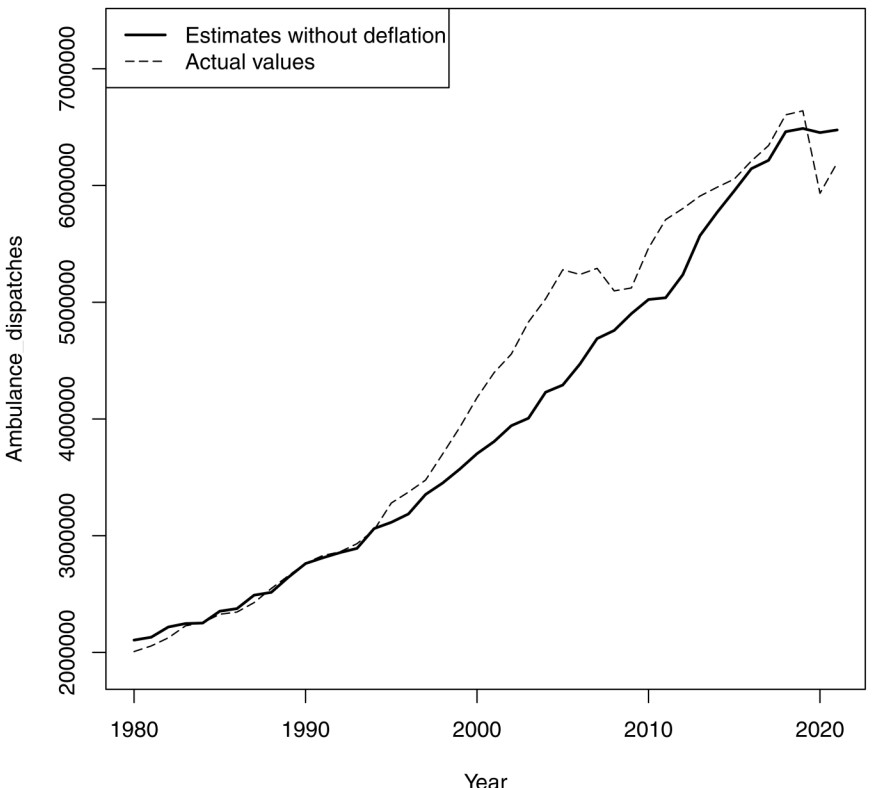

**Fig 3. Ambulance dispatches in Japan and the estimates in case without deflation.** The estimates were recalculated by substituting the regression coefficient of active job openings-to-applicants ratio during inflation for that during deflation in the generalized estimation equation of main analysis to simulate the ambulance dispatch trend in case of absence of deflationary periods.

The sensitivity analysis, which used TOPIX instead of AJOAR, produced similar results. Both AJOAR and stock indices such as TOPIX are recognized as representative macroeconomic indicators, however, they evaluate the macroeconomic environment from different perspectives. The consistent findings from these distinct indicators suggest that overall macroeconomic trends, rather than specific conditions in the labor or stock markets, are linked to the number of ambulance dispatches. Furthermore, these findings validate our hypothesis that deflation is associated with an increase in ambulance dispatches.

The GEE estimate of AJOAR during deflation was determined to be 0.165. If AJOAR×100 were used instead of AJOAR, the estimate would become 0.00165. This corresponds to an annual rate of increase in ambulance dispatches of 1.00165 for every 1% increase in AJOAR, as $e^{0.00165}$ equals 1.00165. Although this rate may appear small, its real-world impact is substantial. For instance, AJOAR increased by 19% between 2003 and 2004. Considering that the number of ambulance dispatches in 2003 was 4,830,813, the expected number of dispatches in 2004 can be calculated as $4,830,813 \times 1.00165^{19} = 4,984,529$. In comparison, the actual number of dispatches in 2004 was 5,029,108, suggesting that approximately 77.5% of the 198,295 increase in dispatches that year was due to the rise in AJOAR.

In the secondary analysis, the GEE coefficient estimate for AJOAR was significant but negative, which contradicted the results of the main analysis. Since the graph did not reflect the actual trend, the negative significance seemed to be a false positive, likely caused by the incorrect assumption that AJOAR consistently influences the number of ambulance dispatches, regardless of inflationary or deflationary periods.

The mean temperature did not seem to be statistically significant when correlated to ambulance dispatches, unlike a previous study that analyzed monthly fluctuations in the number of ambulance dispatches in Japan [12]. This may be associated with using annual data, rather than seasonal data.

## Deflation impact

In a previous longitudinal study by Ruhm covering the period from 1972 to 1991 in the United States, a decrease of one percentage point in the unemployment rate, which indicates temporary economic growth, correlated with approximately a 0.6% increase in resident mortality [15]. The reasons for adverse health effects during such temporary macroeconomic booms were considered to be that people tend to over-work, spend less time taking care of their health, resulting in being prone to traffic accidents in the pursuit of wealth [15]. A longitudinal study in Japan confirmed this theory, demonstrating that an increase in AJOAR led to increased monthly ambulance dispatches from 2003 to 2021 [12]. However, Ruhm also noted that the negative impact of temporary economic booms was counteracted by long-lasting economic growth [15]. The present study revealed similar relationships in the number of ambulance dispatches for the first time.

Considering Ruhm's premise [15], it is possible that long-lasting economic prosperity may mitigate adverse health effects because it reduces the urge to overwork, allowing more time for self-care and leading to fewer work-related accidents. In deflationary periods, a bleak future outlook and accompanying anxiety may lead to increased ambulance dispatches as people prioritize competitive economic opportunities over their health. Conversely, abundant opportunities during inflationary periods may allow people to pursue wealth with a long-term perspective while caring for their health.

Several studies have suggested that anxiety increases ambulance dispatches. A systematic review identified that anxiety, including feeling isolated and having fear of life-threatening conditions, were the driving factors for ambulance use for non-urgent conditions [31]. Furthermore, it signaled that poverty, which amplifies anxiety about the future, was also a factor in non-urgent ambulance service demand [31]. Additionally, a questionnaire survey in Japan with 2,029 participants reported that low income and living alone were associated with non-urgent ambulance use [8]. In another questionnaire survey from the United Kingdom with 2,906 participants, low income and anxiety arising from a sense of powerlessness were reported as factors affecting non-urgent ambulance calls [9]. Consistently, other studies reported that loneliness and low income contributed to the growing demand for ambulance services [7,32]. Deflation, which leads to economic instability, may increase ambulance demand by exacerbating social anxiety.

## Effects of major socioeconomic factors

Our main model did not consider major socioeconomic events such as the economic bubble burst (1995), the global financial crisis (2008), the Great East Japan Earthquake (2011), the introduction of the additional fee for first-time patients without a referral (2016-present), and the COVID-19 pandemic (2020–2022). However, the estimated trend in this model was aligned with published Japanese trend, despite its simple structure. A prior analysis, which examined the monthly ambulance dispatches in Japan from 2003 to 2021 using a similar method demonstrated that dispatch trend could be well explained by AJOAR, without considering other socioeconomic events [12]. These events ultimately affect AJOAR, therefore, AJOAR may account for the trend of ambulance dispatches.

The extended analyses that accounted for major socioeconomic events were also conducted for reference, although the number of explanatory variables exceeded four—one-tenth of the number of outcome points (42 years), which could potentially affect the validity of the GEE analysis. Nevertheless, the significance of AJOAR during deflation remained. Interestingly, only events widely recognized as having a significant impact on nationwide macroeconomics—such as the economic bubble burst, the global financial crisis, and the COVID-19 pandemic [38]—showed a significant negative association. In contrast, localized or individual events unlikely to affect nationwide economic conditions, such as the Great East Japan Earthquake and the introduction of an additional fee for first-time patients without a referral, showed no significant

effects. This finding suggests that large-scale macroeconomic crises may reduce the number of ambulance dispatches, aligning with the results of our main analysis.

## Synergistic effects between AJOAR and long-term macroeconomic status

The graph from the main analysis diverged from the recorded trend in ambulance dispatches around 2013, which coincided with the transition from the prolonged deflationary period to the inflationary period. Additionally, the GEE regression coefficient of AJOAR during deflation differed from that reported in our previous study (0.165 and 0.082, respectively) [12]. These divergences may stem from the oversimplified dichotomous judgement in deflation or inflation. Specifically, when shifting from a deflationary period to an inflationary one with a consistently increasing AJOAR, the regression estimates inevitably show a sharp decline from high to low values. This occurs because AJOAR reaches its peak during deflation just before the transition and then abruptly drops to zero as inflation begins in our model. The degree of AJOAR's contribution may vary depending on macroeconomic phases such as inflation, deflationary transition, deflation, and post-deflation stages. If the contribution of AJOAR closely associated with the macroeconomic phase change, the difference from our prior report could be reconciled. Future studies are required to explore more detailed effects of long-term macroeconomic conditions on ambulance dispatches. We propose here that long-term macroeconomic status has synergistic effects with AJOAR in increasing ambulance dispatches.

## Simulation in case of no deflation

Taking into account the findings of this study, we estimate that with no deflationary periods, about 9.3 million ambulance dispatches between 1995 and 2013 may have been unnecessary in Japan. This may have had a tremendous social and economic impact—less crowded EDs, more timely medical care, and meaningful budget savings. This underscores that implementing societal and economic policies to prevent deflation may be one of the most effective measures in reducing ambulance dispatches. If deflation becomes unavoidable, the government must fundamentally prioritize strengthening and optimizing emergency medical services. However, as this would increase national healthcare expenditures and place additional strain on medical staff, avoiding deflation remains the most desirable outcome. Specific policies, such as measures to prevent deflation, fall outside the scope of this study. Therefore, further research is needed to explore this issue.

## Limitations

This study has several limitations. First, as an ecological study, the findings of this study should not be interpreted as evidence of individual-level causation, a common issue known as the ecological fallacy. Specifically, our results do not indicate that individuals are more likely to call an ambulance when temporary prosperity occurs during deflation. For instance, the increase in ambulance dispatches might have been driven by a specific group of individuals who frequently call an ambulance, while others do not follow this pattern. Moreover, we could not have individual patient characteristics such as age, sex, diseases, and socioeconomic factors because the data had already been aggregated by government offices. Our findings remain valid in the context of population-level trends in Japan.

Second, data were limited to Japan during a specific period, so similar analysis should be conducted in other countries or at different times in the future to assess generalizability of our model. Third, the number of explanatory variables was limited to ensure the reliability of the regression analyses. The variables were carefully selected for valid reasons, as discussed in the "Other factors" section. Nevertheless, there may still be unknown confounding factors beyond those included in this study, and the variable selection may have introduced bias into the results. Fourth, the data of gross prefectural deflator, which corresponds to the GDP deflator, were unavailable before 1995 for each of the Japanese prefectures, not allowing regional subgroup analyses [39]. Fifth, the mean temperature used in the study did not capture seasonal effects. Sixth, the effect of hyperinflation, which brings another type of recession, was not investigated since such a situation did not occur during the covered period. Lastly, lag effects were not considered for simplicity. Future studies are required to address these issues.

## Conclusion

This study demonstrated that the AJOAR during deflation was associated with increased ambulance dispatches. Societal and economic policies to prevent deflation may be essential in order to reduce the number of dispatches. Further research is needed to explore more detailed effects of long-term macroeconomic conditions on ambulance dispatches.

## Supporting information

**S1 Fig. The trend in the number of ambulance dispatches and the deflation periods in Japan.** Arrows indicate major socioeconomic events suffered by Japan of (a) the economic bubble burst, (b) the global financial crisis, (c) the Great East Japan Earthquake, (d) the introduction of the additional fee for a first-time patient without a referral, and (e) the COVID-19 pandemic.
(PDF)

**S1 Table. Comparison of results from the main and sensitivity analyses for annual ambulance dispatches.** GEE: generalized estimation equation; CI: confidence interval; AJOAR: active job openings-to-applicants ratio; TOPIX: Tokyo Stock Price Index; QIC: quasi-likelihood under the independence model criterion. * P < .05. The regression model of main analysis consists of the older population, mean temperature, AJOAR during deflation, and AJOAR during inflation, including the total population as an offset variable. The regression model of sensitivity analysis consists of the older population, mean temperature, TOPIX during deflation, and TOPIX during inflation, including the total population as an offset variable.
(DOCX)

**S2 Fig. Ambulance dispatches in Japan and the regression model of sensitivity analysis.** The regression model consists of the older population, mean temperature, Tokyo Stock Price Index during deflation, and Tokyo Stock Price Index during inflation, including the total population as an offset variable.
(PDF)

**S2 Table. Results of the main analysis and the extended analysis including major socioeconomic events for the annual ambulance dispatches.** GEE: generalized estimation equation; CI: confidence interval; AJOAR: active job openings-to-applicants ratio; QIC: quasi-likelihood under the independence model criterion. * P < .05. The regression model of main analysis consists of the older population, mean temperature, AJOAR during deflation, and AJOAR during inflation, including the total population as an offset variable. The regression model of extended analysis builds on this by additionally including major socioeconomic events: the economic bubble burst, the global financial crisis, the Great East Japan Earthquake, the introduction of the additional fee for a first-time patient without a referral, and the COVID-19 pandemic, while still using the total population as an offset variable.
(DOCX)

**S3 Fig. Ambulance dispatches in Japan and the regression model of extended analysis.** The regression model consists of the older population, mean temperature, active job openings-to-applicants ratio during deflation, active job openings-to-applicants ratio during inflation, the economic bubble burst, the global financial crisis, the Great East Japan Earthquake, the introduction of the additional fee for a first-time patient without a referral, and the COVID-19 pandemic, including the total population as an offset variable.
(PDF)

## Acknowledgments

We would like to thank Professor Ran D. Goldman for his assistance with English language editing and his valuable advice.

## Author contributions

**Conceptualization:** Yohei Kamikawa.

**Data curation:** Yohei Kamikawa.

**Formal analysis:** Yohei Kamikawa.

**Investigation:** Yohei Kamikawa.

**Methodology:** Yohei Kamikawa.

**Project administration:** Yohei Kamikawa.

**Resources:** Yohei Kamikawa.

**Software:** Yohei Kamikawa.

**Supervision:** Yohei Kamikawa.

**Validation:** Yohei Kamikawa, Nao Hanaki.

**Visualization:** Yohei Kamikawa.

**Writing – original draft:** Yohei Kamikawa.

**Writing – review & editing:** Yohei Kamikawa, Nao Hanaki.

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
