## [Decision Letter · Decision Letter 0]

21 Nov 2024

PONE-D-24-37442The changing impact of the active job openings-to-applicants ratio (AJOAR) on ambulance dispatches during deflation: A longitudinal ecological studyPLOS ONE

Dear Dr. Kamikawa,

Thank you for submitting your manuscript to PLOS ONE. After careful consideration, we feel that it has merit but does not fully meet PLOS ONE’s publication criteria as it currently stands. Therefore, we invite you to submit a revised version of the manuscript that addresses the points raised during the review process.

We look forward to receiving your revised manuscript.

Kind regards,

Yusuke Tsutsumi

Academic Editor

PLOS ONE

Journal Requirements:

Reviewers' comments:

Reviewer's Responses to Questions

**Comments to the Author**

1. Is the manuscript technically sound, and do the data support the conclusions?

Reviewer #1: Partly

Reviewer #2: No

Reviewer #3: Yes

Reviewer #4: Yes

2. Has the statistical analysis been performed appropriately and rigorously? 

Reviewer #1: Yes

Reviewer #2: No

Reviewer #3: Yes

Reviewer #4: Yes

3. Have the authors made all data underlying the findings in their manuscript fully available?

Reviewer #1: Yes

Reviewer #2: Yes

Reviewer #3: Yes

Reviewer #4: Yes

4. Is the manuscript presented in an intelligible fashion and written in standard English?

Reviewer #1: Yes

Reviewer #2: Yes

Reviewer #3: Yes

Reviewer #4: Yes

5. Review Comments to the Author

Reviewer #1: Dear Authors,

Thank you for the opportunity to review this intriguing and novel study. Your paper provides valuable insights by analyzing the relationship between macroeconomic indicators and ambulance dispatches from a fresh perspective. However, there are several areas that could benefit from further refinement, particularly in the consideration of explanatory variables, regional differences, and the impact of socioeconomic events. Strengthening the logical explanation of your hypothesis and analysis would significantly enhance the overall quality of the paper.

Below are some key points for your consideration:

1. Explanatory Variables

The use of AJOAR (Active Job Openings-to-Applicants Ratio) as an explanatory variable for ambulance dispatches is somewhat unconventional. While AJOAR is typically employed to explain labor market trends, its causal link to ambulance dispatches remains unclear. The mechanism through which labor market conditions—specifically the balance between job offers and job seekers—impact emergency medical service demand needs further elaboration. Even if there is a hypothesis suggesting that economic instability and stress may indirectly influence health and lead to increased ambulance usage, this connection should be better justified, perhaps by incorporating other macroeconomic factors such as the unemployment rate or GDP growth.

2. Insufficient Consideration of Socioeconomic Events

Significant socioeconomic events such as the bursting of Japan’s bubble economy, the Lehman shock, the Great East Japan Earthquake, and the COVID-19 pandemic likely had considerable impacts on ambulance dispatches. By not accounting for these events, there is a risk of bias in the results. Including these events as adjustment factors could lead to a more robust model. Could you consider incorporating these into your analysis?

3. Regional Differences

While this study utilizes data from across Japan, it is expected that ambulance dispatch patterns and economic conditions differ between urban and rural areas. The exclusion of factors such as regional population density, medical access, and differing attitudes towards emergency medical services may limit the generalizability and accuracy of your model. A subgroup analysis that incorporates regional data could strengthen the robustness of the findings. Is it feasible to attempt such an analysis?

4. Simplistic Treatment of Temperature

The paper uses the annual average temperature as a single explanatory variable, but ambulance dispatches may be heavily influenced by seasonal variations (e.g., the winter flu season). Could you elaborate on the appropriateness of your current method for adjusting for temperature as a confounding factor?

5. Model Explanatory Power

The study employs 42 years of data in its regression analysis, but further discussion is needed regarding the model’s explanatory power. For example, the paper mentions that the model fit deteriorates after 2013, but there is no clear explanation for this. Additionally, while a sensitivity analysis was conducted, the discussion of whether the chosen model is truly optimal remains somewhat vague. Could you clarify these points?

Conclusion

In conclusion, while this paper offers valuable and novel insights, revisions addressing the points above would enhance its overall rigor and impact. I look forward to seeing these improvements in a future version of the manuscript.

Sincerely,

Reviewer

Reviewer #2: Review to the authors in Plos One 2024 Nov

Thank you for the opportunity of reviewing the article on "the active job openings-to-applicants ratio (AJOAR) on ambulance dispatches during deflation: A longitudinal ecological study.” I can understand heavy work load among health staff and try to find the way to decrease ambulance dispatches in Japan. However, there is a critical flaw in an ecological study.

Major comments

1. There are many influential factor to individual outcome (ambulance dispatch in this study), however, an ecological study can not handle these individual-level factors such as age, sex, comorbidity, ADL, and other lifestyles. Thus, employing multivariable analysis with the aggregated data cannot provide any meanings.

Other comments

2. about Table 2.

Line 191 stated that AJOAR varies from 0.63 to 1.09. Regression coefficients of AJOAR (table 2) looks extremely small so that there is almost no relationship with the outcome.

In addition, the coefficient shows opposite (negative) in sensitivity analysis. This means the coefficient is unstable in this study.

3. The author should not refer to non-accepted paper(Research Square).

Reviewer #3: Thank you for giving me an opportunity to check your article titled "The changing impact of the active job openings-to-applicants ratio (AJOAR) on ambulance dispatches during deflation: A longitudinal ecological study". The article is very impressive for ambulance dispatches related to inflation and deflation in Japan.

First of all, the organization of this literature is very clear and makes a lot of sense regarding research design, results, and discussion.Second, We can understand the relationship with economic condition and ambulance dispatches. So, your article is valuable to publish from the journal.

I have some questions.

1. In this period, I know that nationwide pre-hospital telephone consultations and assistance through online medical systems are coming in later in this time setting, but is that impact related to this ambulance mobilization because of pre-hospital consultation services are being established as a policy of the local government?

2. In your conclusions, 'This study demonstrated that the AJOAR during deflation was associated with increased ambulance dispatches.' Are there any specific measures that the government is trying to put in place to address this issue? If you know, please le us know.

3. Regarding the current analysis, the association between economic status and emergency ambulance transportation in an analysis conducted in Japan, a country with a health insurance system in which emergency ambulance transportation is free of charge; would this trend change if the ambulance service were in a country where it is paid for? Also, what do you think will happen if ambulances are paid for in Japan in the future, or if certain standards for ambulance transport are set?

Do you think that the trend of ambulance transports will continue to increase during deflation?

Because if the world economy is moving in the direction of deterioration and is not easily turned toward inflation, do you think there are any measures that can be taken to overcome this situation?

Reviewer #4: Thank you for giving the chance to reviewing this informative research.

I believe this research is extremely important for improving the emergency medical situation.

However, I want to show you some points to be revised before publish.

1. Although authors excluded both individual and rejonal factors from variables, I think the numbers of variables is relatively small. I feel that the number of doctors (especially emergency physicians), emergency institutions, family structure and elderly care facility residents might be related to the number of long term ambulance dispatchs.

2. The authors considered that mean temperature was not a significant relevant variable, which may be due to the use of annual data. I think it is not appropriate to use mean temperature as a variable.

6. PLOS authors have the option to publish the peer review history of their article (what does this mean? ). If published, this will include your full peer review and any attached files.

**Do you want your identity to be public for this peer review?** For information about this choice, including consent withdrawal, please see our Privacy Policy .

Reviewer #1: **Yes: ** Kiyomitsu Fukaguchi

Reviewer #2: No

Reviewer #3: **Yes: ** Fumihiro Ogawa

Reviewer #4: No

---

## [Author Response · Author response to Decision Letter 0]

31 Dec 2024

Dr. Emily Chenette,

Editor-in-Chief

PLOS ONE

Dear Dr. Chenette,

We would like to thank you and the reviewers for your consideration and comments on our manuscript. We have revised the manuscript accordingly and have provided a point-by-point response to the comments below.

We hope that our manuscript will now be suitable for publication in your journal.

Kindest regards,

Yohei Kamikawa

Department of Emergency Medicine, University of Fukui Hospital, Fukui, Japan

E-mail: fundarike.ykami@gmail.com

Dear Editor,

Thank you very much for your careful consideration of our article.

Question: Please ensure that your manuscript meets PLOS ONE's style requirements, including those for file naming. The PLOS ONE style templates can be found at

Answer: We agree that our manuscript should comply with the requirements. We have carefully reviewed the templates and revised the format accordingly (Lines 12, 14, 31, 32, 62, 68, 69, 81, 135, 156, 201, 202, 215, 291, 292, 323, 355, 379-380, 399, 413, 435, 442, 446, 536).

Dear Reviewer #1,

Thank you very much for your careful consideration of our article.

Question: 1. Explanatory Variables

The use of AJOAR (Active Job Openings-to-Applicants Ratio) as an explanatory variable for ambulance dispatches is somewhat unconventional. While AJOAR is typically employed to explain labor market trends, its causal link to ambulance dispatches remains unclear. The mechanism through which labor market conditions—specifically the balance between job offers and job seekers—impact emergency medical service demand needs further elaboration. Even if there is a hypothesis suggesting that economic instability and stress may indirectly influence health and lead to increased ambulance usage, this connection should be better justified, perhaps by incorporating other macroeconomic factors such as the unemployment rate or GDP growth.

Answer: We agree that additional macroeconomic indicators should be examined to confirm that macroeconomic trends influence the number of ambulance dispatches. Therefore, we conducted sensitivity analyses by replacing AJOAR with TOPIX, a stock price index, in the main analyses. The results were consistent with those of the main analyses. Consequently, as different macroeconomic indicators from varying perspectives yielded similar results, it can be concluded that macroeconomic trends—rather than the conditions of the labor or stock markets—are associated with the number of ambulance dispatches. These arguments have been added to the manuscript (Lines 92-93, 167-169, 209-210, 212-213, 224-230, 251-252, 293, 295-301, S1 Table, S2 Fig). The discussion on how macroeconomic trends influence the number of dispatches is already included in the subsection "Deflation impact" within the Discussion section, with references to previous studies (Lines 323-353).

Question: 2. Insufficient Consideration of Socioeconomic Events

Significant socioeconomic events such as the bursting of Japan’s bubble economy, the Lehman shock, the Great East Japan Earthquake, and the COVID-19 pandemic likely had considerable impacts on ambulance dispatches. By not accounting for these events, there is a risk of bias in the results. Including these events as adjustment factors could lead to a more robust model. Could you consider incorporating these into your analysis?

Answer: We acknowledge that incorporating socioeconomic events could potentially lead to more robust conclusions. However, when the number of explanatory variables exceeds four—one-tenth of the total number of outcome points (42 years)—the validity of the GEE analysis may be compromised. Therefore, we performed the analyses as an extended analysis for reference. As a result, only events with nationwide economic impact were found to be statistically significant alongside AJOAR during deflation. These points have been added to the manuscript (Lines 185-189, 280-289, 366-377, S2 Table, S3 Fig).

Question: 3. Regional Differences

While this study utilizes data from across Japan, it is expected that ambulance dispatch patterns and economic conditions differ between urban and rural areas. The exclusion of factors such as regional population density, medical access, and differing attitudes towards emergency medical services may limit the generalizability and accuracy of your model. A subgroup analysis that incorporates regional data could strengthen the robustness of the findings. Is it feasible to attempt such an analysis?

Answer: We acknowledge that subgroup analyses incorporating regional data could have provided more robust conclusions. However, as mentioned in the Limitations section (Lines 422-424), such analyses were not feasible due to the unavailability of gross prefectural deflator data prior to 1995. Nevertheless, this may limit the generalizability of the study. Therefore, we added a statement in the Limitations section to acknowledge the possibility of unknown factors beyond the variables examined in our model, which may have influenced the results (Lines 419-420).

Question: 4. Simplistic Treatment of Temperature

The paper uses the annual average temperature as a single explanatory variable, but ambulance dispatches may be heavily influenced by seasonal variations (e.g., the winter flu season). Could you elaborate on the appropriateness of your current method for adjusting for temperature as a confounding factor?

Answer: We acknowledge that the annual mean temperature may not capture seasonal variations. However, this study follows the methodology of a previous study that demonstrated a significant impact of mean temperature on the number of ambulance dispatches. Excluding this variable from our model would have been overly arbitrary, so we decided to retain it. These considerations have been added to the Methods section (Lines 111-116).

Question: 5. Model Explanatory Power

The study employs 42 years of data in its regression analysis, but further discussion is needed regarding the model’s explanatory power. For example, the paper mentions that the model fit deteriorates after 2013, but there is no clear explanation for this. Additionally, while a sensitivity analysis was conducted, the discussion of whether the chosen model is truly optimal remains somewhat vague. Could you clarify these points?

Answer: We agree that the explanatory power of the models should be discussed further. To address this, we added Goodness-of-Fit tests to the statistical analyses. These tests confirmed that there were no significant issues with the model’s fit. This information has been incorporated into the manuscript (Lines 165-167, 222-223, 228-229, 232-233, 286-287).

Regarding the apparent poor fit around 2013, we consider that this is due to the oversimplified dichotomous classification of periods as either deflationary or inflationary, as described in the manuscript (Lines 385-386). Specifically, during the transition from deflation to inflation, when the AJOAR consistently increases, the regression estimates in our model show a sharp decline. This occurs because AJOAR peaks just before the transition during deflation and then abruptly drops to zero at the onset of inflation. This explains the deviation observed around 2013. We have added this explanation to the Discussion section (Lines 386-390).

As for whether our model is truly optimal, we acknowledge the possibility of unknown variables and more refined models. Since our model was based on a previous study, it may not represent the most optimal approach. We have added this consideration to the manuscript (Lines 114-116, 419-420).

Dear Reviewer #2,

Thank you very much for your careful consideration of our article.

Question: 1. There are many influential factor to individual outcome (ambulance dispatch in this study), however, an ecological study can not handle these individual-level factors such as age, sex, comorbidity, ADL, and other lifestyles. Thus, employing multivariable analysis with the aggregated data cannot provide any meanings.

Answer: We acknowledge that analyzing individual-level factors would be ideal. However, such analyses were not possible due to the fully anonymized nature of the data provided by government offices. As a result, we used aggregated data to capture general population trends in Japan, which allowed us to obtain meaningful and beneficial findings. This discussion has been added to the Limitations section (Lines 426-427, 428-430).

With regard to multivariable analyses using aggregated data, we have previously conducted similar analyses, and the results were published in a peer-reviewed journal:

Reference: Kamikawa Y. Impact of the active job openings-to-applicants ratio on the number of ambulance dispatches in Japan, 2003–2021: a longitudinal ecological study. BMJ Open. 2024;14:e083755.

Additionally, since numerous studies have employed multivariable analyses with aggregated data, we believe there are no methodological issues with our approach. Examples include:

Reference: Yazaki H, Nishiura H. Ambulance Transport of Patients with Mild Conditions in Hokkaido, Japan. Int J Environ Res Public Health. 2020 Feb 2;17(3):919.

Reference: Ikeda N, Fuse K, Nishi N. Changes in the effects of living with no siblings or living with grandparents on overweight and obesity in children: Results from a national cohort study in Japan. PLoS One. 2017 Apr 17;12(4):e0175726.

Question: 2. about Table 2.

Line 191 stated that AJOAR varies from 0.63 to 1.09. Regression coefficients of AJOAR (table 2) looks extremely small so that there is almost no relationship with the outcome.

In addition, the coefficient shows opposite (negative) in sensitivity analysis. This means the coefficient is unstable in this study.

Answer: While the GEE coefficient estimate of 0.165 may appear small, its real-world impact is substantial. We have added this clarification to the Discussion section (Lines 303-312).

Regarding the term “sensitivity analysis,” we acknowledge that this terminology was inappropriate. The purpose of these analyses was not to evaluate the robustness of the model but rather to compare the fit between our current model and a previous study’s model. Therefore, the term “secondary analysis” would have been more accurate. We have revised this terminology throughout the manuscript.

Considering this clarification, the coefficient estimate of AJOAR during deflation was not unstable. The secondary analyses were not designed to test minor variations in conditions but rather to assess a fundamental change in the hypothesis—specifically, whether the contribution of AJOAR varies between inflationary and deflationary periods. For example, if we were investigating whether a plant releases more oxygen or carbon dioxide, the result might indicate less oxygen depending on the observation period. However, if we were testing the hypothesis that a plant’s gas emissions differ between daytime and nighttime, we could find that it releases more oxygen during the day. Should we conclude that the amount of oxygen released is unstable because of a negative result in the secondary analysis? The answer is “no.” In this case, we should conclude that the hypothesis of constant oxygen and carbon dioxide emissions is incorrect, and instead, a plant produces more oxygen during the day and less at night. We hope this analogy helps clarify that our study employed a similar type of analysis, focusing on testing changes in the underlying hypothesis.

The significant negative estimate for the GEE coefficient in the secondary analysis was likely a misleading result caused by a fundamental flaw in the hypothesis that the contribution of AJOAR remained constant regardless of whether the period was inflationary or deflationary. The p-value is known to be unreliable when the hypothesis is incorrect.

Reference: Price R, Bethune R, Massey L. Problem with p values: why p values do not tell you if your treatment is likely to work. Postgrad Med J. 2020 Jan;96(1131):1-3. doi: 10.1136/postgradmedj-2019-137079. Epub 2019 Oct 29. PMID: 31662411.

Since relying solely on p-values can lead to misinterpretation, additional evidence is necessary to avoid such errors. In this study, we included graphical confirmation to validate the findings. The graph from the main analysis aligned well with the actual trend, whereas the graph from the secondary analysis failed to reflect the observed trend. Based on this, we could conclude that AJOAR contributed to the number of ambulance dispatches only during deflationary periods. These points have been added to the Discussion section (Lines 313-317).

Question: 3. The author should not refer to non-accepted paper(Research Square).

Answer: Fortunately, the paper was recently accepted and published in BMJ Open. We have added it to the References section (Lines 472-474).

Dear Reviewer #3,

Thank you very much for your careful consideration of our article.

Question: 1. In this period, I know that nationwide pre-hospital telephone consultations and assistance through online medical systems are coming in later in this time setting, but is that impact related to this ambulance mobilization because of pre-hospital consultation services are being established as a policy of the local government?

Answer: We recognize that pre-hospital telephone consultations and online medical assistance systems may influence the number of ambulance dispatches. However, some municipalities have yet to implement these services. Even among those that have, the timing of implementation varies. Therefore, accounting for such factors is not appropriate in this study, which focuses on nationwide trends affecting dispatch numbers. We acknowledge that conducting subgroup analyses incorporating regional data could have strengthened our conclusions. However, as noted in the Limitations section (Lines 422-424), such analyses were not feasible due to the unavailability of gross prefectural deflator data before 1995. Nevertheless, this limitation may affect the generalizability of our findings. To address this, we added a statement in the Limitations section acknowledging the potential influence of unknown factors not examined in this study (Lines 419-420).

Question: 2. In your conclusions, 'This study demonstrated that the AJOAR during deflation was associated with increased ambulance dispatches.' Are there any specific measures that the government is trying to put in place to address this issue? If you know, please le us know.

Answer: Some argue that Japan's deflation, which began in 1995, was driven by inappropriate monetary and fiscal policies that contradicted economic principles, and that proper policies aligned with contemporary economic theories could have prevented it. However, we believe that exploring which policies might mitigate deflation falls outside the scope of this study. This research focuses exclusively on the relationship between ambulance dispatch numbers and macroeconomic indicators. Therefore, we have refrained from commenting on matters not directly supported by our findings. This perspective is addressed in the Discussion section (Lines 409–411). If your question pertains to what the government should prioritize during deflation, the focus must be on strengthening and optimizing emergency medical services. That said, the exploration of specific responses to such scenarios also lies beyond the scope of this study. Accordingly, we have deliberately avoided discussing issues not substantiated by our results. This point has been also highlighted in the Discussion section (Lines 406–411).

Question: 3. Regarding the current analysis, the association between economic status and emergency ambulance transportation in an analysis conducted in Japan, a country with a health insurance system in which emergency ambulance transportation is free of charge; would this trend change if the ambulance service were in a country where it is paid for? Also, what do you think will happen if ambulances are paid for in Japan in the future, or if certain standards for ambulance transport are set?

Do you think that the tre

---

## [Decision Letter · Decision Letter 1]

15 Jan 2025

PONE-D-24-37442R1The changing impact of the active job openings-to-applicants ratio (AJOAR) on ambulance dispatches during deflation: A longitudinal ecological studyPLOS ONE

Dear Dr. Kamikawa,

Thank you for submitting your manuscript to PLOS ONE. After careful consideration, we feel that it has merit but does not fully meet PLOS ONE’s publication criteria as it currently stands. Therefore, we invite you to submit a revised version of the manuscript that addresses the points raised during the review process.

We look forward to receiving your revised manuscript.

Kind regards,

Yusuke Tsutsumi

Academic Editor

PLOS ONE

**Additional Editor Comments:**

Thank you for the opportunity to handling this study. Basically, I acknowledge the clinical importance of this study.

However, after checking the reviewers' comments and the authors' responses, I, the editor, find there are still major points that need to be addressed.

Major points

1. While I acknowledge that ecological study is an established study design that provides a certain level of meaningful evidence, as the authors have responded, I also recognize that this design has substantial limitations, such as ecological fallacy, which aligns with Reviewer 2's concerns. Therefore, I strongly believe the authors need to add appropriate discussion about the limitations inherent to the study design.

2. I agree with Reviewer 4's point to the initial submission regarding the relatively small number of variables. There may be residual confounding, and the results could be biased. Therefore, the authors should discuss how this limited number of variables might affect the results and what aspects readers should consider when interpreting them. In the current form, the authors state: "Second, the number of explanatory variables was limited in order to maintain the reliability of the regression analyses... Nevertheless, this study effectively captured the trend in ambulance dispatches despite its concise model structure." While this description acknowledges the limitation, the authors claim the results are not problematic without providing adequate rationale or discussion.

3. Regarding the fundamental methodology, including variable selection, the validity of "mean" temperature, and the validity of the simple model (with few variables), the authors have cited their own previous study (reference 11). While I acknowledge that reference 11 is clinically important, for methodological issues, the authors should provide additional references from other researchers demonstrating that these methods are widely accepted and valid, such as methodological studies or similar research papers.

Minor point

4. The authors have stated that they could not obtain individual patient characteristics "due to the fully anonymized data provided by government offices." I believe the precise reason is that the data was already aggregated by government offices rather than being fully anonymized, which would suggest that the data was individual but anonymized.

Reviewers' comments:

Reviewer's Responses to Questions

**Comments to the Author**

1. If the authors have adequately addressed your comments raised in a previous round of review and you feel that this manuscript is now acceptable for publication, you may indicate that here to bypass the “Comments to the Author” section, enter your conflict of interest statement in the “Confidential to Editor” section, and submit your "Accept" recommendation.

Reviewer #1: All comments have been addressed

Reviewer #2: (No Response)

Reviewer #3: All comments have been addressed

2. Is the manuscript technically sound, and do the data support the conclusions?

Reviewer #1: Yes

Reviewer #2: No

Reviewer #3: Yes

3. Has the statistical analysis been performed appropriately and rigorously? 

Reviewer #1: Yes

Reviewer #2: No

Reviewer #3: Yes

4. Have the authors made all data underlying the findings in their manuscript fully available?

Reviewer #1: Yes

Reviewer #2: Yes

Reviewer #3: Yes

5. Is the manuscript presented in an intelligible fashion and written in standard English?

Reviewer #1: Yes

Reviewer #2: Yes

Reviewer #3: Yes

6. Review Comments to the Author

Reviewer #1: Dear Authors,

Thank you for your thorough and thoughtful revisions. I appreciate your efforts to address the concerns raised during the review process, especially the inclusion of sensitivity analyses using TOPIX alongside AJOAR. This addition significantly strengthens the robustness of your findings by demonstrating consistency across different macroeconomic indicators.

Final Suggestions:

To further enhance the clarity and impact of your manuscript, I suggest emphasizing the significance of the TOPIX sensitivity analyses in the Discussion section. Explicitly highlighting how the consistent results validate the role of macroeconomic trends would provide additional support for your conclusions.

Your work offers insights into the connection between macroeconomic conditions and ambulance dispatch trends, with meaningful implications for public health. I believe your manuscript is nearly ready for acceptance with this minor revision.

Sincerely,

Reviewer #2: Review to the authors in Plos One 2025 Jan

Thank you for the second opportunity of reviewing the article on "the active job openings-to-applicants ratio (AJOAR) on ambulance dispatches during deflation: A longitudinal ecological study”. I can understand the answer for question 2 of mine in revised article, however, an ecological study cannot detect risk factors of ambulance dispatches.

Major comments

An ecological study can easily contain an ecological fallacy. As authors stated in line 141-145 in the revised manuscript, individual factors such as health status, comorbidities, and other health-related lifestyles, are indispensable for the multivariable regression.

AJOAR, older population, and low-temperature cannot fully explain annual ambulance.

Kamikawa’s past paper in BMJ Open should take additional review if possible.

Moreover, Ikeda et al.’s paper handled the individual-level analysis, not aggravated data.

The authors must obtain individual-level health data.

Reviewer #3: Thank you so much for reply to our comments to you about the article. My doubts have been resolved. I have also checked the comments of other reviewers, but they all make sense and I have nothing more to say. This has made things very clear.

7. PLOS authors have the option to publish the peer review history of their article (what does this mean? ). If published, this will include your full peer review and any attached files.

**Do you want your identity to be public for this peer review?** For information about this choice, including consent withdrawal, please see our Privacy Policy .

Reviewer #1: **Yes: ** Kiyomitsu Fukaguchi

Reviewer #2: No

Reviewer #3: No

---

## [Author Response · Author response to Decision Letter 1]

24 Feb 2025

Dr. Emily Chenette,

Editor-in-Chief

PLOS ONE

Dear Dr. Chenette,

We would like to thank you and the reviewers for your consideration and comments on our manuscript. We have revised the manuscript accordingly and have provided a point-by-point response to the comments below.

We hope that our manuscript will now be suitable for publication in your journal.

Kindest regards,

Yohei Kamikawa

Department of Emergency Medicine, University of Fukui Hospital, Fukui, Japan

E-mail: fundarike.ykami@gmail.com

Dear Editor,

Thank you very much for your careful consideration of our article.

Question: 1. While I acknowledge that ecological study is an established study design that provides a certain level of meaningful evidence, as the authors have responded, I also recognize that this design has substantial limitations, such as ecological fallacy, which aligns with Reviewer 2's concerns. Therefore, I strongly believe the authors need to add appropriate discussion about the limitations inherent to the study design.

Answer: We agree that the limitations related to the study design should be discussed in more detail. In the Limitations section, we have added a caution regarding the ecological fallacy, emphasizing that the results should not be interpreted as evidence of individual-level causation but remain valid in the context of population-level trends. (Lines 422-431)

Question: 2. I agree with Reviewer 4's point to the initial submission regarding the relatively small number of variables. There may be residual confounding, and the results could be biased. Therefore, the authors should discuss how this limited number of variables might affect the results and what aspects readers should consider when interpreting them. In the current form, the authors state: "Second, the number of explanatory variables was limited in order to maintain the reliability of the regression analyses... Nevertheless, this study effectively captured the trend in ambulance dispatches despite its concise model structure." While this description acknowledges the limitation, the authors claim the results are not problematic without providing adequate rationale or discussion.

Answer: We agree that the limitations due to the small number of variables should be discussed. First, we removed any wording that could be misinterpreted as implying that the results were free of issues. While we have already explained the rationale for variable selection in the “Other factors” section (Lines 135-159), we have now clarified the possibility of residual confounding and variable selection bias in the Limitations section. (Lines 435-438)

Question: 3. Regarding the fundamental methodology, including variable selection, the validity of "mean" temperature, and the validity of the simple model (with few variables), the authors have cited their own previous study (reference 11). While I acknowledge that reference 11 is clinically important, for methodological issues, the authors should provide additional references from other researchers demonstrating that these methods are widely accepted and valid, such as methodological studies or similar research papers.

Answer: We agree that a more detailed methodological discussion is necessary.

Regarding variable selection, we have added a methodological article on ecological studies as reference 26. This article emphasizes the importance of selecting appropriately aligned group-level variables when analyzing group-level trends in ecological studies. Excluding macro-level variables could overlook crucial factors, such as trends in macroeconomic indicators that undeniably influence broader patterns. From this perspective, AJOAR, the older population, and mean temperature are suitable variables for analyzing national annual trends in ambulance dispatches, as each exhibits its own national annual trend. Conversely, the absence of individual-level data is not a major concern in an ecological study. Applying individual-level inference to group-level trends may not be appropriate, as theories relevant at the individual level do not necessarily hold at the population level. We have incorporated this reference and expanded the explanation in the Methods section to clarify the rationale behind our variable selection. (Lines 136-139, 519-520)

Regarding the validity of mean temperature, we have already cited reference 10, which demonstrated an association between mean temperature and the number of ambulance dispatches. This study, conducted in China from 2015 to 2016, found that ambulance dispatches were lowest at a mean temperature of 22.5°C and increased as the temperature deviated from that point, forming a U-shaped relationship. It also showed that replacing mean temperature with minimum or maximum temperature yielded similar results. Additionally, we have added another reference (reference 11) that further supports the association between mean temperature and the number of ambulance dispatches. (Lines 43, 154, 483-485)

One key reason for the simplicity of our model is that, among the variables previously reported to increase ambulance dispatches, only AJOAR, the older population, and mean temperature were national annual group-level variables that corresponded with national annual ambulance dispatch trends. As discussed above and stated in the Methods section, these were the most appropriate variables for inclusion (Lines 136-139). Additionally, it is generally accepted that the number of explanatory variables should be limited to one-tenth of the outcome data points to ensure the reliability of regression analysis. While this point had already been mentioned in the Methods section, we have now added a reference that explains the rationale behind this limitation (Lines 156, 542-547). Additionally, previous studies have also employed simple models. For instance, Wang et al. (reference 10) performed a regression analysis using only mean temperature as an explanatory variable in their main analysis without considering any confounders, though they conducted subgroup analyses based on gender, age group, and initial diagnosis category. An et al. (reference 11) incorporated mean temperature while adjusting for confounding factors such as seasonal trends, meteorological conditions, air pollutant levels, and day of the week, but did not account for demographic characteristics. Compared to these studies, our model incorporates a broader range of variables. We have clarified this point in the Methods section. (Line 154)

As discussed, we have added references and further discussion to clarify the validity of our methodology. We hope this addresses your concerns.

Question: 4. The authors have stated that they could not obtain individual patient characteristics "due to the fully anonymized data provided by government offices." I believe the precise reason is that the data was already aggregated by government offices rather than being fully anonymized, which would suggest that the data was individual but anonymized.

Answer: We agree with your suggestion. The section containing the sentence has been revised as recommended. (Lines 429-430)

Dear Reviewer #1,

Thank you very much for your careful consideration of our article.

Question: To further enhance the clarity and impact of your manuscript, I suggest emphasizing the significance of the TOPIX sensitivity analyses in the Discussion section. Explicitly highlighting how the consistent results validate the role of macroeconomic trends would provide additional support for your conclusions.

Answer: We appreciate your suggestion. To highlight the importance of the sensitivity analysis, we created a dedicated paragraph and clarified that our hypothesis—that deflation leads to an increase in ambulance dispatches—was validated. (Lines 303-310)

Dear Reviewer #2,

Thank you very much for your careful consideration of our article.

Question: An ecological study can easily contain an ecological fallacy.

Answer: We agree that the limitations related to the study design should be discussed in more detail. In the Limitations section, we have added a caution regarding the ecological fallacy, emphasizing that the results should not be interpreted as evidence of individual-level causation but remain valid in the context of population-level trends. (Lines 422-431)

Question: As authors stated in line 141-145 in the revised manuscript, individual factors such as health status, comorbidities, and other health-related lifestyles, are indispensable for the multivariable regression.

AJOAR, older population, and low-temperature cannot fully explain annual ambulance.

Kamikawa’s past paper in BMJ Open should take additional review if possible.

Moreover, Ikeda et al.’s paper handled the individual-level analysis, not aggravated data.

The authors must obtain individual-level health data.

Answer: In an ecological study, individual-level data are not always essential. This study does not aim to investigate why “individuals” call an ambulance but rather to examine the factors driving the “national annual” trend in ambulance dispatches. When analyzing such group-level tendencies, the absence of individual-level data is not a major concern. Instead, the key is to select appropriately aligned group-level variables.

From this perspective, AJOAR, the older population, and mean temperature are suitable variables, as each exhibits its own national annual trend. Conversely, applying individual-level inference to group-level trends may not be appropriate, as theories relevant at the individual level do not always hold at the population level.

Moreover, excluding macro-level variables would overlook crucial factors—such as trends in macroeconomic indicators—that are difficult to capture through individual-level data, even in economics, yet undeniably influence broader trends.

Technically, it is also impossible to analyze national annual trends in ambulance dispatches using only individual-level data, as such an analysis would require data from Japan's entire population of approximately 120 million people over a span of 42 years.

These points are explained in detail by Zeoli et al. Therefore, we have added this article as Reference 26 and provided further explanations in the Methods section to clarify the validity of our variable selection. (Lines 136-139, 519-520)

If you are looking for additional research utilizing aggregated data without using individual-level data, please refer to the following articles.

Reference: Ishikawa M, Hemmi O, Takimoto H, Matsumoto M, Yokoyama T. Trend of estimated participation rate by regional block, gender, and age group in the 1997-2019: National Health and Nutrition Survey in Japan. PLoS One. 2024 Mar 13;19(3):e0286169.

Reference: Hamashima C, Sano H. Association between age factors and strategies for promoting participation in gastric and colorectal cancer screenings. BMC Cancer. 2018 Mar 27;18(1):345.

Reference: Matsumoto Y, Nakai A, Nishijima Y, Kishita E, Hakuno H, Sakoi M, Kusuda S, Unno N, Tamura M, Fujii T. Absence of neonatal intensive care units in secondary medical care zones is an independent risk factor of high perinatal mortality in Japan. J Obstet Gynaecol Res. 2016 Oct;42(10):1304-1309.

Dear Reviewer #3,

Thank you very much for your careful consideration of our article. We understand that there are no further questions, and therefore, no additional response is necessary.

---

## [Editor Report · Decision Letter 2]

27 Feb 2025

The changing impact of the active job openings-to-applicants ratio (AJOAR) on ambulance dispatches during deflation: A longitudinal ecological study

PONE-D-24-37442R2

Dear Dr. Kamikawa,

We’re pleased to inform you that your manuscript has been judged scientifically suitable for publication and will be formally accepted for publication once it meets all outstanding technical requirements.

Kind regards,

Yusuke Tsutsumi

Academic Editor

PLOS ONE

Additional Editor Comments (optional):

I see the authors have responded appropriately to the remarks made by the editor and reviewers. As long as the limitations are acknowledged appropriately, I judge the ecological study can provide a meaningful level of evidence.
---

## [Editor Report · Acceptance letter]

PONE-D-24-37442R2

PLOS ONE

Dear Dr. Kamikawa,

I'm pleased to inform you that your manuscript has been deemed suitable for publication in PLOS ONE. Congratulations! Your manuscript is now being handed over to our production team.

Kind regards,

on behalf of

Dr. Yusuke Tsutsumi

Academic Editor

PLOS ONE